# An interpretable framework for investigating the neighborhood effect in POI recommendation

**Guangchao Yuan**[1], **Munindar P. Singh**[2], **Pradeep K. Murukannaiah**[3]*

**1** Microsoft, Mountain View, CA, United States of America, **2** North Carolina State University, Raleigh, NC, United States of America, **3** Delft University of Technology, Delft, The Netherlands

* P.K.Murukannaiah@tudelft.nl

**Data Availability Statement:** All data supporting the findings in this paper are openly available at https://doi.org/10.5281/zenodo.4489876.

**Funding:** There was no specific funding for our work. "The first author (Guangchao Yuan) is

## Abstract

Geographical characteristics have been proven to be effective in improving the quality of point-of-interest (POI) recommendation. However, existing works on POI recommendation focus on cost (time or money) of travel for a user. An important geographical aspect that has not been studied adequately is the *neighborhood effect*, which captures a user's POI visiting behavior based on the user's preference not only to a POI, but also to the POI's neighborhood. To provide an interpretable framework to fully study the neighborhood effect, first, we develop different sets of insightful features, representing different aspects of neighborhood effect. We employ a Yelp data set to evaluate how different aspects of the neighborhood effect affect a user's POI visiting behavior. Second, we propose a deep learning–based recommendation framework that exploits the neighborhood effect. Experimental results show that our approach is more effective than two state-of-the-art matrix factorization–based POI recommendation techniques.

## 1 Introduction

Location-based social networks (LBSNs) facilitate personalized point-of-interest (POI) recommendation, which not only helps users in exploring interesting places, but also increases revenues for businesses. Several POI recommendation approaches employ a user-POI check-in matrix, which is typically very sparse (most users usually check-in only at a few POIs). Accordingly, approaches that rely only on the check-in matrix yield poor recommendation quality [1]. Additional context information such as social relationships [2] and geographical characteristics [3] have been exploited to overcome the data sparsity problem and improve the recommendation quality.

The neighborhood of a POI consists of its surrounding POIs. We define the *neighborhood effect* as a phenomenon that a user visits a POI because of their preference not only to that POI but also to the POI's neighborhood. The neighborhood effect is different from the geographical effect investigated in previous studies. Those studies assume that users tend to visit POIs near their homes or offices and nearby POIs of a POI that they just visited [4, 5]. The geographical effect is mainly about the cost of travel (e.g., time or money) from a user's current location to a

employed by Microsoft. Microsoft provided support in the form of salaries for Guangchao Yuan, but did not have any additional role in the study design, data collection and analysis, decision to publish, or preparation of the manuscript. The specific role of the author is articulated in the 'author contributions' section. This does not alter our adherence to PLOS ONE policies on sharing data and materials." Please update this information as necessary.

**Competing interests:** The first author (Guangchao Yuan) is employed by Microsoft. Microsoft provided support in the form of salaries for Guangchao Yuan, but did not have any additional role in the study design, data collection and analysis, decision to publish, or preparation of the manuscript. The specific role of the author is articulated in the 'author contributions' section. This does not alter our adherence to PLOS ONE policies on sharing data and materials.

POI. In contrast, the neighborhood effect is about the environmental context created by a POI's surrounding POIs. For example, a user is more likely to visit a restaurant that is surrounded by their interested POIs than an isolated restaurant, although the two restaurants are at the same distance from the user's current location. In this case, the geographical effect is the same for the two restaurants, but the the neighborhood effect affects the user's behavior.

The neighborhood effect has not been studied adequately in POI recommendation. Liu et al. [6] find that the neighborhood effect is more important than the geographical effect in POI recommendation. Similarly, Li et al. [7] integrate the neighborhood effect into a matrix factorization–based model. Doan et al. [8] investigate two main properties of the neighborhood effect, area attraction and neighborhood competition, and ascertain the two factors through empirical studies. However, most studies (1) obtain the latent vectors of users and POIs from the matrix factorization model, limiting the *interpretability* of the neighborhood effect; and (2) measure the neighborhood effect purely from a personalized effect perspective, provided that each POI in a neighborhood plays a role in a user's preference for the neighborhood—the inner product of latent vectors between a user and a POI measures a user's preference for the POI, and summation of such preferences from a user to each POI in a neighborhood is the neighborhood's attraction to the user. What if such preference is more dominated by a single POI (e.g., the least or most preferred POI)? What if the environmental context of a neighborhood is the key factor that attracts a user? For example, a user with kids may visit a highly rated kids-friendly plaza, even though none of the POIs in the neighborhood attracts the user individually. Yin et al. [9] propose several sets of features to model the environmental context, but they ignore the personalized preference.

To address the shortcomings above, we develop an interpretable POI recommendation framework in which the neighborhood effect can be *fully investigated* and *explicitly evaluated*. In particular, we seek to answer two research questions.

**RQ$_1$: Neighborhood Effect Modeling** How to model the neighborhood effect in an interpretable and comprehensive manner? In particular, what constitutes the neighborhood effect and how it affects a user's POI visiting behavior?

**RQ$_2$: POI Recommendation** How to build a POI recommender that exploits the neighborhood effect to improve recommendation quality?

To answer these questions, we analyze data from three cities—Phoenix, Toronto, and Las Vegas—with the highest number of POIs in the Yelp Challenge data set [10].

To answer RQ$_1$ on neighborhood modeling, we factorize the neighborhood effect into two main categories: (1) *personalized* features capture a user's personalized preference over nearby POIs; and (2) *property* features model the environmental context of a neighborhood. We further divide *personalized* features into four subsets by employing different aggregating functions—two subsets aggregate a user's personalized preference over each nearby POI by arithmetic (distance adapted) mean functions, and the other two subsets model the neighborhood effect via maximum (minimum) functions. Our evaluation results from the first research question indicate that (1) the neighborhood effect could benefit POI recommendation; (2) there is no significant difference between the *property* features and *personalized* features; and (3) there is no significant difference among different aggregating functions in terms of modeling a user's personalized preference for a neighborhood.

To answer RQ$_2$ on POI recommendation, we propose a multilayer perceptron (MLP)–based recommender, incorporating the neighborhood effect. Our model outperforms two state-of-the-art matrix factorization–based approaches, demonstrating the effectiveness of integrating neighborhood features in a deep neural network for POI recommendation.

Although our approach only integrates the neighborhood effect, it can be extended to other contexts (e.g., social and temporal influences). It is important for a recommender to integrate different kinds of context to improve its quality. However, some previous approaches are developed for a specific kind of context [1, 11], and they do not generalize.

## 1.1 Key contributions

- We quantify different aspects of the neighborhood effect and evaluate the potential benefits of each aspect, providing an interpretable framework for understanding the neighborhood effect. To the best of our knowledge, our study is the first to comprehensively investigate and evaluate different aspects of the neighborhood effect.

- We propose a deep neural network–based POI recommender that incorporates the neighborhood effect and improves precision and recall of POI recommendation compared to the state-of-the-art matrix factorization models. The proposed recommender can be easily extended to incorporate other context information.

## 2 Related work

We review POI recommendation approaches that exploit geographical characteristics and works that are closely related to our deep learning–based recommendation approach.

## 2.1 POI recommendation

Geographical characteristics are usually studied from the users' perspective (*geographical effect*) or from the locations' perspective (*neighborhood effect*). Most of the existing works focus on the geographical effect [1, 2, 12].

A few studies focus on the geographical characteristics from the locations' perspective. Several researchers integrate the neighborhood effect into a matrix factorization–based model [6, 7], showing the effectiveness of the neighborhood effect. Yin et al. [9] propose several sets of features to model the environmental context created by the neighborhood. Doan et al. [8] investigate two main properties of the neighborhood effect: area attraction and neighborhood competition. Most existing studies either assume that the neighborhood effect is from a user's personalized preferences for the nearby POIs [6, 8] or from the environmental context [9]. However, the neighborhood effect may include both of these aspects. We investigate neighborhood effect in a comprehensive and interpretable manner compared to the existing works.

Zhang and Chow [1] emphasize that the category of a POI captures the function of a POI, and a user's visiting behavior at a POI reflects their interests in the corresponding category. Accordingly, our approach incorporates the POI category information in defining the relevant features.

## 2.2 Deep learning–based recommender systems

There is an increasing interest in deep learning–based POI recommenders. Due to the sequence visiting behavior seen for POIs, some researchers build models based on recurrent neural networks (RNN) [13, 14]. However, these approaches cannot be applied to scenarios where sequence behavior is not available. Yin et al. [15] build a convolutional neural network–based model to extract features from users' check-ins. Due to widespread availability, some researchers exploit content information to build deep learning–based models for recommendation [16, 17]. Most of these works model the check-in behavior or content information via

deep neural models. In contrast, our work focuses on integrating the heterogeneous information learned from feature engineering with deep neural networks. Due to the performance boosting capability of word embedding in the NLP field, many researchers focus on learning POI embedding through deep learning models [18]. Xie et al. [19] jointly capture different context in a unified way by embedding the four corresponding relation graphs into a low dimensional space. Instead of learning a latent vector that could better represent a POI, our work focuses more on investigating the neighborhood properties.

He et al.'s approach [20] is most relevant to ours. They explore neural network architectures for collaborative filtering using only the check-in matrix. They provide a general architecture to model the user-item interactions in three different ways. Covington et al. [21] describe a practical deep neural network–based recommendation architecture for YouTube. They also model the user-item interactions through multiple layers of fully connected Rectified Linear Units. We adopt a similar architecture to model the user-item interactions, but focus on modeling neighborhood effect, and integrating the effect with deep neural models for recommendation.

## 3 Problem formulation

Let $U = \{u\}_{u=1}^{M}$ be a set of $M$ users and $L = \{l\}_{l=1}^{N}$ be a set of $N$ POIs. Each POI is associated with a geo-tag and a POI category. A user visits one or more POIs. Some users also have online friendships with other users. Then, we formulate our research questions as follows.

RQ$_1$: **Neighborhood Effect Modeling** Given a user $u$'s observed check-ins and the neighborhood $L_n^{-l}$ for each of their checked-in POI $l$, how can we model the neighborhood effect in an interpretable and comprehensive manner?

That is, how can we factorize the neighborhood effect into two feature vectors $\omega_u$ and $\omega_l$? The first feature vector, $\omega_u$, embeds the user and the neighborhood pair, representing the user's personalized preference for the neighborhood. The second feature vector, $\omega_l$, embeds the POI and the neighborhood pair, representing the characteristics of the neighborhood the POI belongs to.

RQ$_2$: **POI Recommendation** Given the neighborhood effect ($\omega_u$ and $\omega_l$), how can we predict the user $u$'s preference score $\hat{y}_{u,l}$ for an unvisited POI $l$ and return the top-$k$ POIs with the highest scores?

$$\hat{y}_{u,l} = f(v_u, v_l, \omega_u, \omega_l; P, Q, \Theta_f) \tag{1}$$

where $P \in \mathbb{R}^{M \times K}$ and $Q \in \mathbb{R}^{N \times K}$ are the latent factor matrices for users and POIs, respectively; $K$ is the dimension of the latent space; $v_u \in \mathbb{R}^{M \times 1}$ and $v_l \in \mathbb{R}^{N \times 1}$ are the sparse one-hot encoding vectors for user $u$ and POI $l$, respectively; $\Theta_f$ denotes the parameters for the model function $f$.

We answer these questions via the Yelp Challenge data set [10]. We conduct experiments on three cities with the highest number of POIs in the data set. The three cities (Phoenix, Toronto, and Las Vegas) represent different life styles. Phoenix and Toronto represent daily-life styles, where several check-ins are likely from residents. However, Toronto is more densely populated than Phoenix. Las Vegas represents tourism style, where several check-ins are likely from tourists.

The Yelp data set does not contain check-in frequencies—it only provides users' reviews of POIs. Therefore, we only know whether a user has visited that POI but not the exact number if the user visited the POI more than once.

**Table 1. Statistics from the Yelp data set for three cities.**

| City | Users | POIs | Reviews | #POIs per user | #Users per POI |
|---|---|---|---|---|---|
| Phoenix | 3,566 | 13,154 | 157,189 | 44.07 | 11.95 |
| Toronto | 3,811 | 16,042 | 207,800 | 54.53 | 12.95 |
| Las Vegas | 10,606 | 23,816 | 490,730 | 46.27 | 20.61 |

We clean the data set by removing users with fewer than 20 reviews. Table 1 reports the statistics of the data set in each city after cleaning. Each POI is associated with a geo-tag (latitude, longitude) and a top-level POI category [22]. The resulting data set covers POIs from 21 different categories. The data set also contains social relationships between users. That is, for each user, it provides a list of the user's friends on Yelp.

## 4 Neighborhood effect modeling (RQ₁)

Fig 1 (top) plots the percentage of POIs having at least one other POI within 100 m, 200 m, or 500 m distance. We calculate the geographical distance between two POIs by the Haversine formula [23]. When the threshold is 200 m, ~95% of POIs in each of the three cities have at least one neighbor.

We define the size of a neighborhood as the number of POIs in its neighborhood. We adopt the threshold of 200 m to define a neighborhood. That is, the neighborhood of a POI is given by the set of POIs within 200 m radius of that POI. We adopt this definition of a neighborhood due to its simplicity and popularity among researchers [6, 24], but our study is not limited to this definition. With the threshold of 200 m, almost all POIs have some neighbor but not too many POIs have too many neighbors. Fig 1 (bottom) outlines the frequency of neighborhood sizes with the threshold of 200 m in each city. We observe that, the majority of neighborhoods have fewer than 20 POIs, and only a few neighborhoods consist of than 100 POIs.

Our goal is to model the neighborhood effect when a user decides to visit a POI. We seek to provide an interpretable framework to fully investigate the neighborhood effect by answering questions such as: What constitutes the neighborhood effect and what is the corresponding performance on POI recommendation? Is the neighborhood effect mainly dominated by a user's personalized preferences for the nearby POIs or the neighborhood property? Is it more affected by a single POI or all the nearby POIs? Is it dominated by the least or most preferred POI?

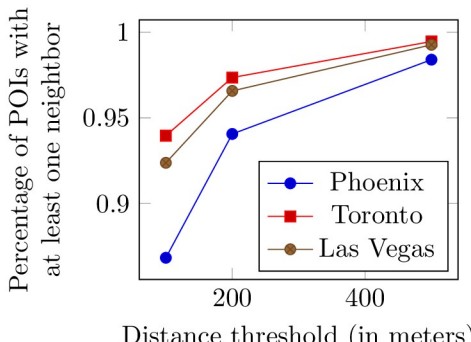 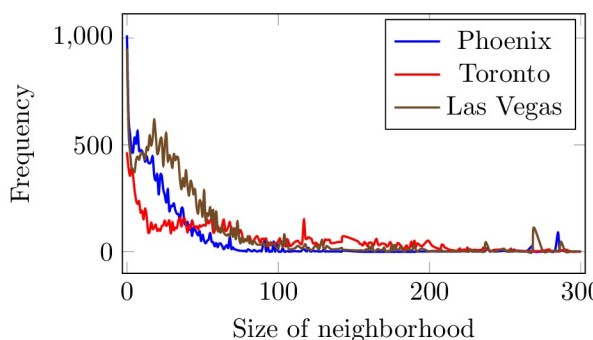

**Fig 1.** The percentage of POIs having at least one neighbor within a distance threshold of 100 m, 200 m, and 500 m (top). The frequency of different neighborhood sizes in each city, considering neighborhood size of 200 m (bottom).

**Table 2. Notation.**

| | |
|---|---|
| $n_{u,l}$ | The number of times a user $u$ visits POI $l$ |
| $n_{u,c}$ | The number of times a user $u$ visits POIs belonging to category $c$ |
| $r_{u,l}$ | The rating a user $u$ gives to a POI $l$ |
| $n_l$ | The number of times a POI $l$ has been visited |
| $U_l$ | The set of users who have visited a POI $l$ |
| $U_c$ | The set of users who have visited POIs belonging to category $c$ |
| $C_l$ | The set of categories POI $l$ belongs to |
| $L_n^{-l}$ | The set of POIs, excluding POI $l$, that are located in the neighborhood of $l$ |
| $L_{n,c}^{-l}$ | The set of POIs belonging to category $c$, excluding POI $l$, located in the neighborhood of $l$ |
| $U_n^{-l}$ | The set of users who have visited at least one POI in the neighborhood of $l$, not including POI $l$ |

In Sections 4.1 and 4.2, we define features to model a user's preference toward a single POI (*user-POI* features) and for a neighborhood (*user-neighborhood* features), respectively. These fine-grained features help in interpreting the neighborhood effect and answering the questions above. Table 2 shows the notation we adopt.

## 4.1 User-POI features

Instead of learning latent vector of users and POIs from matrix factorization models [6, 7] or deep learning–based models [18, 19], we propose several sets of *user-POI* features to represent a user's preference for a POI for its better interpretability. In addition, most matrix factorization–based methods learn the latent vectors from the check-in matrix, which is usually very sparse. We exploit different sources of information to overcome the data sparsity problem.

We divide *user-POI* features into five categories.

**4.1.1 Check-in features (RATING-FACTORIZATION).** Due to the lack of check-in frequency data, we exploit matrix factorization [25] on the observed user-POI rating matrix to factorize users and POIs into a joint latent factor space. The inner products between a user and a POI's latent factors in this space measures how much a user is interested in the POI.

**4.1.2 Content features (CONTENT-EMBEDDING).** Word embedding [26] maps words from a high dimensional space to a continuous vector space of a much lower dimension such that words with similar meaning have a similar vector representation. Recently, this approach has gained popularity due to its performance boosting capability in many content analysis applications [17].

We train the word embedding for reviews in each city separately using the word2vec algorithm [26]. Before training, we preprocess each review by converting all words into lower case and removing all stop words, punctuations, and non-alphabetical words. Next, we retain only words that appear at least four, four, and nine times, resulting in a vocabulary size of 42,212, 45,881, and 50,601 in Phoenix, Toronto, and Las Vegas, respectively. For Phoenix and Toronto, the threshold of four reduces the vocabulary size to 20% of the original vocabulary size. We choose this threshold by following an assumption (based on the Pareto principle [27]) that 20% of vocabulary can cover 80% of the user preferences expressed in the reviews. We choose a different threshold for Las Vegas compared to the other two cities because the vocabulary size in Las Vegas is much larger than the other two cities we study. We choose nine as the threshold for Las Vegas because it yields a vocabulary size that resembles the other two cities.

For each user, we aggregate all their reviews and average the embedding of all the words, resulting in the user embedding. For each POI, we aggregate all the reviews of the POI and

average the embedding of all the words, resulting in the POI embedding. We average instead of concatenating embeddings by following Covington et al. [21]. As for the check-in features, the inner products between user and POI embedding measures how likely the user will visit the POI.

**4.1.3 Popularity features.** A POI's popularity is an important indicator of user interests in the POI [28]. We define popularity by three metrics:

(1). the number of visits,

$$\text{NUMBER-VISITS-POI} = \sum_{u \in U_l} n_{u,l} = n_l, \tag{2}$$

(2). the average rating of the POI,

$$\text{AVERAGE-RATING-POI} = (\sum_{u \in U_l} r_{u,l})/n_l = r_l, \tag{3}$$

(3). the number of users who have visited the POI,

$$\text{NUMBER-USERS-POI} = |U_l|. \tag{4}$$

In our Yelp data set, $n_{u,l}$ is always one. However, we define this feature so that it is applicable to other data sets that contain user check-in frequency. The popularity of a POI cannot distinguish one user's preference for a POI from another user. In contrast, the POI category can capture personalized preference because users may show distinct biases (or affinities) over POI categories, e.g., a user will go to bars if they are interested in *socializing* and to gyms if they are interested in *fitness*.

We define a user's bias for a category $c$ as

$$b_{u,c} = \frac{n_{u,c}}{\sum_{c \in C} n_{u,c}}. \tag{5}$$

In Eq 5, $C$ is the predefined set of categories in the data set. For users with insufficient history for a category, we introduce a parameter $\tau$ to overcome "cold start" [29]. That is, $b_{u,c} = \tau \times d_c$, where $d_c$ represents the category density in one city—the number of POIs belonging to category $c$ divided by the number of POIs in a city. We set $\tau$ to 0.3 in our experiment.

Following Zhang et al. [1], we measure a user's preference for a POI based on the POI's popularity weighted by the user's category bias. A POI may be associated with several categories. We define three popularity features:

(1). the categorical popularity by number of visits,

$$\text{VISIT-POPULARITY} = \sum_{c \in C_l} b_{u,c} \times \frac{n_l}{\sum_{l \in C} n_l}, \tag{6}$$

(2). the categorical popularity by rating,

$$\text{RATING-POPULARITY} = \sum_{c \in C_l} b_{u,c} \times \frac{r_l}{\sum_{l \in C} r_l}, \qquad (7)$$

(3). the categorical popularity by number of users,

$$\text{USER-POPULARITY} = \sum_{c \in C_l} b_{u,c} \times \frac{|U_l|}{\sum_{l \in C} |U_l|}. \qquad (8)$$

**4.1.4 Geographical features.** Due to the importance of geographical influence in a user's location visiting behavior, we also need to consider the distance between the location of POI $l$ and their visited locations. For each user, we compute a *centroid* location $m_u$, where the latitude and longitude is the mean values of the latitudes and longitudes of their visited geo-coordinates. We define two geographical features:

(1). the median of distances between POI $l$ and each visited POI of user $u$ (MEDIAN-DISTANCE), and

(2). the distance between POI $l$ and $m_u$ (DISTANCE-TO-CENTROID).

**4.1.5 Social features.** We include two social features:

(1). the sum of visits over each user $u$'s friends for POI $l$ (SOCIAL-VISIT), and

(2). the average of ratings over each user $u$'s friends for POI $l$ (SOCIAL-RATING).

## 4.2 User-neighborhood features

For a user $u$ and a POI $l$, we define $u$'s preference for $l$'s neighborhood by excluding $u$'s preference for $l$ so that we can investigate how $l$'s nearby POIs affect $u$'s preference for $l$.

Intuitively, the neighborhood effect may comprise two factors: the user's preference toward each POI in the neighborhood, individually (*personalized* features) and the properties of the neighborhood (*property* features). According to the problem formulation in Section 3, *personalized* features correspond to $\omega_u$ and *property* features correspond to $\omega_l$.

**4.2.1 Personalized features.** We divide *personalized* features into four subsets by employing different aggregating functions—two subsets aggregate a user's personalized preference over each nearby POI by arithmetic (distance adapted) mean functions, and the other two subsets model the neighborhood effect via maximum and minimum functions. Previous studies [6–8] model the neighborhood effect via the arithmetic (distance adapted) mean functions. To the best of our knowledge, no other researchers have investigated the performance of the neighborhood effect via maximum and minimum functions.

To simplify the notation, let $\delta_{u,j}$ represent one of the *user-POI* features. Thus, each one of the following subsets includes nine features in evaluation.

(1). Average preference over each nearby POI

$$\text{AVERAGE-SCORE} = (\sum_{j \in L_n^{-l}} \delta_{u,j})/|L_n^{-l}|. \tag{9}$$

(2). Average preference over each nearby POI, but the similarity is adjusted by the distance,

$$\text{DISTANCE-SCORE} = (\sum_{j \in L_n^{-l}} \delta_{u,j} \times s_{l,j})/(\sum_{j \in L_n^{-l}} s_{l,j}). \tag{10}$$

Following [6], we exploit a Gaussian distribution over the distance between POI $l$ and POI $j$ to model their similarity. That is,

$$s_{l,j} = e^{-\frac{(d_{l,j})^2}{(\sigma)^2}}, \tag{11}$$

where $d_{l,j}$ is the geographical distance between POI $l$ and POI $j$ and $\sigma$ is a constant, which we set as 0.01 in our experiment.

(3). Maximum preference among nearby POIs,

$$\text{MAX-SCORE} = \max_{j \in L_n^{-l}} \delta_{u,j}. \tag{12}$$

(4). Minimum preference among nearby POIs,

$$\text{MIN-SCORE} = \min_{j \in L_n^{-l}} \delta_{u,j}. \tag{13}$$

**4.2.2 Property features.** *Property* features capture the environmental context created by the POIs in a neighborhood. For example, is the neighborhood a *food plaza* or a *shopping mall*? Is it popular? We define a neighborhood's popularity on a category $c$ as the proportion of POIs belonging to category $c$:

$$\rho_{n,c}^{-l} = \frac{|L_{n,c}^{-l}|}{|L_n^{-l}|} \tag{14}$$

We employ $\rho_{n,c}^{-l}$ in multiple features below.

(1). Number of POIs in the neighborhood,

$$\text{NUMBER-OF-POI} = |L_n^{-l}|. \tag{15}$$

(2). We apply the entropy measurement to the neighborhood's popularity over a category $c$:

$$\text{CATEGORY-ENTROPY} = -\sum_{c \in C} \rho_{n,c}^{-l} \times \log \rho_{n,c}^{-l}. \tag{16}$$

(3). For each user, we obtain their category-bias vector, where each item in the vector is the user's bias for the category $c$: $b_{u,c}$. For each neighborhood, we obtain its category distribution vector, where each item is the neighborhood's popularity over the category $\rho_{n,c}^{-l}$. Then, we compute the cosine similarity between the two vectors:

$$\text{CATEGORY-SIMILARITY} = \frac{\sum_{c \in C} b_{u,c} \times \rho_{n,c}^{-l}}{\sqrt{\sum_{c \in C} (b_{u,c})^2} \times \sqrt{\sum_{c \in C} (\rho_{n,c}^{-l})^2}}. \tag{17}$$

(4). This feature measures how many other POIs in the same neighborhood having the same category with POI $l$. The competitiveness may have both positive and negative effects on a user's POI visiting behavior. For some categories like "nightlife", a higher number of POIs having the same category may be more attractive to users. However, for categories like "restaurants", users may become more selective when deciding which POI to visit, reducing the chances of visiting some of them. Because a POI could have several categories $C_l$, we pick the category that has maximum neighborhood popularity:

$$\text{COMPETITIVENESS} = \max_{c \in C_l} \rho_{n,c}^{-l}. \tag{18}$$

(5). Sum of visits over all other POIs,

$$\text{NEIGHBORHOOD-VISIT} = \sum_{j \in L_n^{-l}} n_j. \tag{19}$$

(6). Average of ratings over all other POIs,

$$\text{NEIGHBORHOOD-RATING} = \left(\sum_{j \in L_n^{-l}} r_j\right) / |L_n^{-l}|. \tag{20}$$

(7). Number of unique users who visited at least one another POI in the neighborhood,

$$\text{NEIGHBORHOOD-USER} = |U_n^{-l}|. \tag{21}$$

## 5 Evaluation of neighborhood effect

We seek to answer two questions: (1) Does the neighborhood effect benefit POI recommendation? (2) What is the performance of different ways of modeling the neighborhood effect on POI recommendation?

### 5.1 Evaluation strategy

For each user, we randomly choose 80% of their visited POIs as the training set, and the rest of visited POIs as the test data. In addition, for each user, we sample a set of unvisited POIs whose number is the same with their visited POIs, and 80% of them serve as the training data. In this way, we transform the recommendation problem to a binary classification problem. A

**Table 3. AUC and accuracy for different classifiers on different combination of features.**

| | City | Metric | POI | Neighborhood | Property | Personalized | | | |
|---|---|---|---|---|---|---|---|---|---|
| | | | | | | Average | Distance | Max | Min |
| Logistic Regression | Phoenix | AUC | 0.8319 | **0.8581** | 0.8333 | 0.8553 | 0.8552 | 0.8535 | 0.8563 |
| | | Accuracy | 0.7540 | **0.7824** | 0.7565 | 0.7733 | 0.7731 | 0.7729 | 0.7739 |
| | Toronto | AUC | 0.8077 | **0.8256** | 0.8076 | 0.8201 | 0.8200 | 0.8204 | 0.8221 |
| | | Accuracy | 0.7353 | **0.7478** | 0.7353 | 0.7427 | 0.7427 | 0.7432 | 0.7443 |
| | Las Vegas | AUC | 0.8560 | **0.8683** | 0.8569 | 0.8647 | 0.8646 | 0.8621 | 0.8649 |
| | | Accuracy | 0.7731 | **0.7896** | 0.7764 | 0.7856 | 0.7853 | 0.7836 | 0.7847 |
| FastTree | Phoenix | AUC | 0.8788 | **0.8819** | 0.8809 | 0.8796 | 0.8795 | 0.8795 | 0.8800 |
| | | Accuracy | 0.8115 | **0.8135** | 0.8132 | 0.8123 | 0.8118 | 0.8123 | 0.8126 |
| | Toronto | AUC | 0.8585 | **0.8681** | 0.8672 | 0.8608 | 0.8607 | 0.8608 | 0.8595 |
| | | Accuracy | 0.7862 | **0.7942** | 0.7941 | 0.7880 | 0.7885 | 0.7872 | 0.7877 |
| | Las Vegas | AUC | 0.8958 | **0.9002** | 0.8999 | 0.8967 | 0.8971 | 0.8963 | 0.8971 |
| | | Accuracy | 0.8239 | **0.8279** | 0.8274 | 0.8258 | 0.8257 | 0.8253 | 0.8248 |

pair is constructed between a user and a POI, and the label of the pair is positive if the user has visited the POI. The features of each pair are computed based on the knowledge from the training data. We apply logistic regression and a MART gradient boosting algorithm *FastTree* [30]. All the features are normalized to [0, 1] before evaluation.

## 5.2 Evaluation results

We adopt the *area under the curve* (AUC) and *accuracy* as the classification evaluation metrics. Table 3 compares the classification metrics with and without neighborhood features. We highlight the highest value in each row. For simplicity, we use "POI" to represent *user-POI* features, and "neighborhood" to represent all (*personalized* and *property*) *user-neighborhood* features.

Because *user-POI* features play a major role in POI recommendation, all classifiers include those features. For example, "property" represents that *user-POI* features and *property* features are included in the corresponding classifier.

First, adding *user-neighborhood* features increases AUC and accuracy in all three cities for both classifiers. This suggests that the neighborhood effect can benefit POI recommendation.

Second, the improvement that *user-neighborhood* features yield is more obvious with logistic regression than for FastTree. We conjecture that *user-neighborhood* features overcome the simplicity of a model (as in logistic regression); thus, when the model becomes more sophisticated (as in FastTree, which can recognize complex decision boundaries), the improvement from *user-neighborhood* features is not as pronounced.

Next, we evaluate each type of *user-neighborhood* features to further investigate the neighborhood effect.

(1). If we compare *property* features with each subset of *personalized* features, the AUC and accuracy of the *personalized* features are better than the *property* features for Logistic Regression. However, for FastTree, *property* features alone yield similar AUC and accuracy as the combination of *property* and *personalized* features. Thus, our evaluation results do not show any significant difference between *property* features and *personalized* features.

(2). There is not much difference among the subsets of *personalized* features in terms of AUC and accuracy. If we consider all the nearby POIs in a neighborhood, there is not much

difference if we adjust the similarity by the distance (average vs. distance). For the maximum (minimum) functions, because taking the maximum (minimum) of each *user-POI* feature would result in different POIs, we further investigate the impact of each *user-POI* feature.

For each classifier, we take all the *user-POI* features for POI *l* and maximum (minimum) of a specific *user-POI* feature over all the nearby POIs. Table 4 reports AUC for FastTree classifier on each user-POI feature with maximum (minimum) aggregating functions. For example, "visit-popularity" with the "max" function represents all the *user-POI* features for POI *l* and the maximum of "visit-popularity" feature among all the other POIs in the neighborhood. We choose FastTree because it has a better performance than logistic regression.

(1). Our evaluation results do not show any significant difference between maximum and minimum aggregating functions.

(2). In Table 4, we highlight the top-2 ranked feature with aggregation function in terms of AUC across each city. The *rating-popularity* feature with the maximum function always ranks within the top-2 among the three cities. Our results show that *rating-popularity* is an effective feature in modeling the neighborhood effect: the maximum value of the single feature shows a comparable performance with all the personalized features over all the nearby POIs shown in Table 3.

## 6 Exploiting neighborhood features for POI recommendation (RQ$_2$)

We showed different ways of modeling neighborhood effect and their corresponding performance on POI recommendation. Next, we propose **NbdMLP**, an MLP-based recommender that exploits the neighborhood features (described in Section 4).

**Table 4. AUC for FastTree on each user-POI feature with maximum (minimum) aggregating functions.**

| Feature | Function | Phoenix | Toronto | Las Vegas |
|---|---|---|---|---|
| Rating-factorization | Max | 0.8790 | 0.8586 | 0.8959 |
|  | Min | 0.8790 | 0.8587 | 0.8959 |
| Content-embedding | Max | 0.8790 | **0.8596** | **0.8965** |
|  | Min | 0.8794 | 0.8590 | 0.8963 |
| Visit-popularity | Max | 0.8792 | 0.8595 | 0.8964 |
|  | Min | 0.8791 | 0.8589 | 0.8961 |
| Rating-popularity | Max | **0.8796** | **0.8596** | **0.8965** |
|  | Min | 0.8794 | 0.8593 | 0.8964 |
| User-popularity | Max | **0.8798** | **0.8598** | 0.8963 |
|  | Min | 0.8793 | 0.8595 | 0.8964 |
| Median-distance | Max | 0.8788 | 0.8587 | 0.8958 |
|  | Min | 0.8790 | 0.8588 | 0.8957 |
| Distance-to-centroid | Max | 0.8785 | 0.8588 | 0.8953 |
|  | Min | 0.8789 | 0.8590 | 0.8959 |
| Social-visit | Max | 0.8789 | 0.8584 | 0.8956 |
|  | Min | 0.8787 | 0.8584 | 0.8960 |
| Social-rating | Max | 0.8782 | 0.8584 | 0.8956 |
|  | Min | 0.8790 | 0.8584 | 0.8960 |

## 6.1 NbdMLP model architecture

An advantage of deep neural network–based recommenders (compared to traditional matrix factorization) is the convenience of integrating heterogeneous, e.g., continuous and categorical, features [21]. Previous studies have shown that an MLP-based model could mimic collaborative filtering and yield a comparable performance [20, 21]. We adopt a similar architecture due to its simplicity, effectiveness of imitating collaborative filtering, and convenience of integrating heterogeneous features.

Since we choose the multilayer neural network as the function $f$ in Eq 1, the equation can be formulated as

$$\hat{y}_{u,l} = \phi_{out}(\phi_X(\ldots(\phi_2(\phi_1(P^T v_u, Q^T v_l, \omega_u, \omega_l))\ldots))), \tag{22}$$

where $\phi_{out}$ is the mapping function for the output layer and $\phi_x$ represents the mapping function for the $x$-th layer, where $x \in [1, 2, \ldots, X]$.

Fig 2 shows the NbdMLP model architecture. The input layers resemble the check-in matrix, with a difference that each user and POI are transformed into a sparse vector with one-hot encoding. The sparse input vector of a user (a POI) is embedded into a dense vector. The obtained embedding of a user (a POI) layer can be seen as a latent vector in the collaborative filtering. Importantly, the embeddings are learned jointly with other parameters in the model training process.

The user and the POI embedding are concatenated with the neighborhood features, producing a wide first layer of an MLP architecture. Inside MLP layers, several layers are fully connected, and we choose Rectifier (ReLU) [31] as the activation function. The final output layer is the predicted score $\hat{y}_{u,l}$, and the training objective is to minimize the pointwise loss between $\hat{y}_{u,l}$ and its target label $y_{u,l}$ (0 or 1).

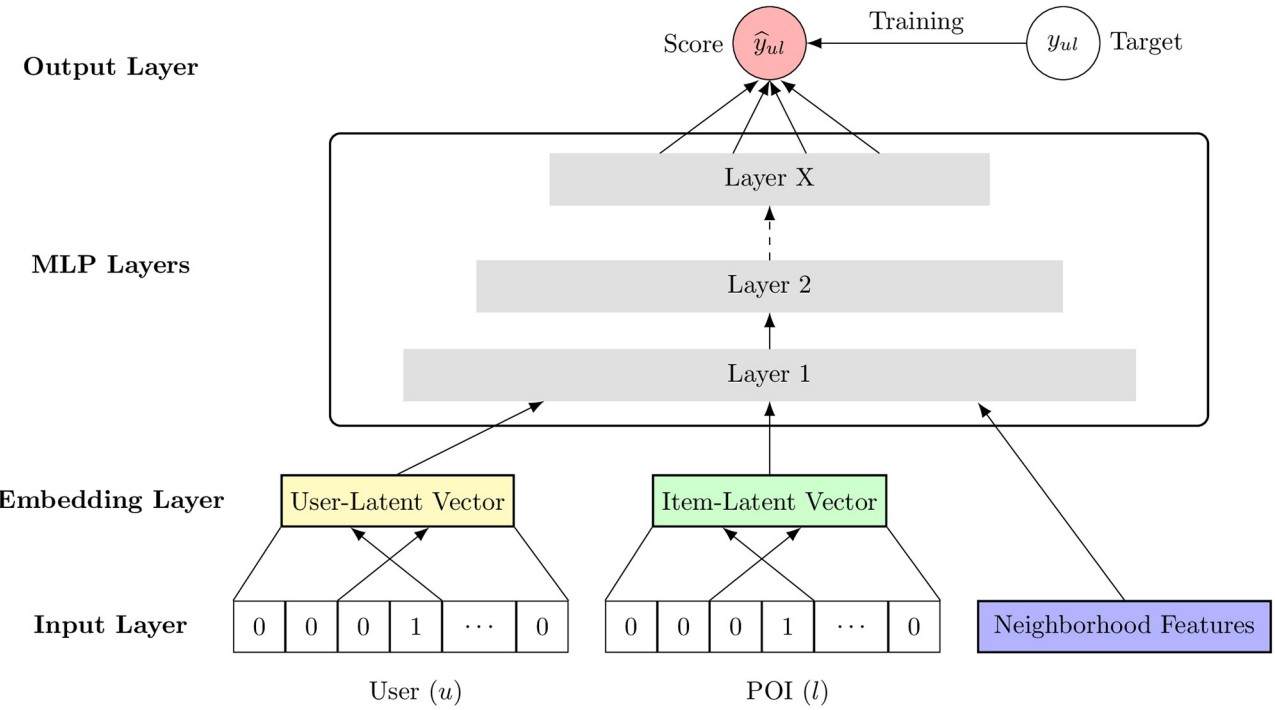

**Fig 2. The architecture of the NbdMLP model.**

Because *property* features perform in par with *personalized* features (Section 5), we focus only on *property* features with the user and POI embeddings for simplicity. We normalize all such features to [0, 1] before feeding to the network.

The user and POI embedding can be pretrained with other models to improve the recommender's quality, which is not the focus of this study. The purpose of our work is to investigate different aspects of neighborhood effect and propose a recommender that could easily integrate the neighborhood features.

## 6.2 Evaluation strategy and metrics

For each user, we randomly choose 80% of their visited POIs for training, 10% of visited POIs for validation (to tune parameters), and the rest of visited POIs for testing.

To solve the one-class CF problems [32], for each observed check-in pair, we randomly sample five unobserved user-POI pairs as negative instances. As He et al. [20] suggest, the optimal sampling ratio is around 3 to 6.

It is time consuming to rank all unvisited POIs for each user. Thus, as Koren's [33] work, we randomly sample 100 POIs that a user has not visited and add these to the test set of the user. Then, we rank the POIs in this combined set of visited and unvisited POIs. We treat a POI as *discovered* if it is visited and recommended. We apply the same strategy to all the baseline methods. Based on this strategy, we evaluate the POI recommendation quality via two metrics:

- Precision@N (P@N) = Number of discovered POIs among the top-N POIs / Number of recommended POIs

- Recall@N (R@N) = Number of discovered POIs among the top-N POIs / Number of actual visited POIs

Specifically, we report P@5, R@5, P@10, and R@10.

We implement *NbdMLP* with Keras [34]. In deep neural models, more hidden layers and more neurons in each layer would improve the model performance, but may require significantly better server configurations. We select hidden layer parameters, considering both efficiency and effectiveness. First, we choose the user and POI embedding size as 16. Because the first layer in the MLP layers is concatenation of user and POI embeddings and neighborhood features, its size is fixed. After experiments with different hidden layers on the validation set, we pick 32 and 16 ReLU units in the second and third layers, respectively. We discuss more details of hidden layer configurations in Section 6.5. We adopt batch size of 256, epochs of 10, learning rate of 0.001, and Adaptive Moment Estimation (Adam) [35] as the optimizer.

## 6.3 Baseline recommendation approaches

We compare our approach, NbdMLP, with the two state-of-the-art recommendation algorithms and their variants. To make these approaches comparable with NbdMLP, we select the latent factors of users and POIs as 16, and other parameters based on the best performance on the validation set.

**LIBMF** is the standard matrix factorization technique [36], where only the check-in matrix is used for recommendation. We set the L2-norm regularization on users and POIs as 0.1, 0.05, respectively. The number of iterations is 10.

**IRenMF** is a state-of-the-art matrix factorization–based approach exploiting neighborhood effect [6]. It models neighborhood effect both at the instance level (a few nearest POIs) and at the region level (the geographical region where the POI is located). We set the

regularization parameters $\lambda_1$ and $\lambda_2$ to 0.015, and the instance weighting parameter $\alpha$ to 0.6. We choose the k-means algorithm to cluster all POIs into 100 regions. For each POI, the closest 10 POIs are selected to model its instance-level neighborhood.

**InMF** is a special case of IRenMF, where the neighborhood effect only exploits the instance-level characteristics.

**RenMF** is a special case of IRenMF, where the neighborhood effect only exploits the region-level characteristics.

**MLP** is a special case of *NbdMLP*, without the neighborhood features. We adopt this baseline to understand how the neighborhood features affect recommendation. We set the parameters of MLP to be same as those for NbdMLP.

## 6.4 Recommendation performance comparison

Fig 3 compares the recommendation metrics. First, compared with the other matrix factorization–based baselines (all baselines except MLP), NbdMLP shows the best performance for both P@5 and R@5 across the three cities. For P@10 and R@10, NbdMLP outperforms other baselines except LIBMF on Toronto, with around 0.002 decrease for both metrics. Considering

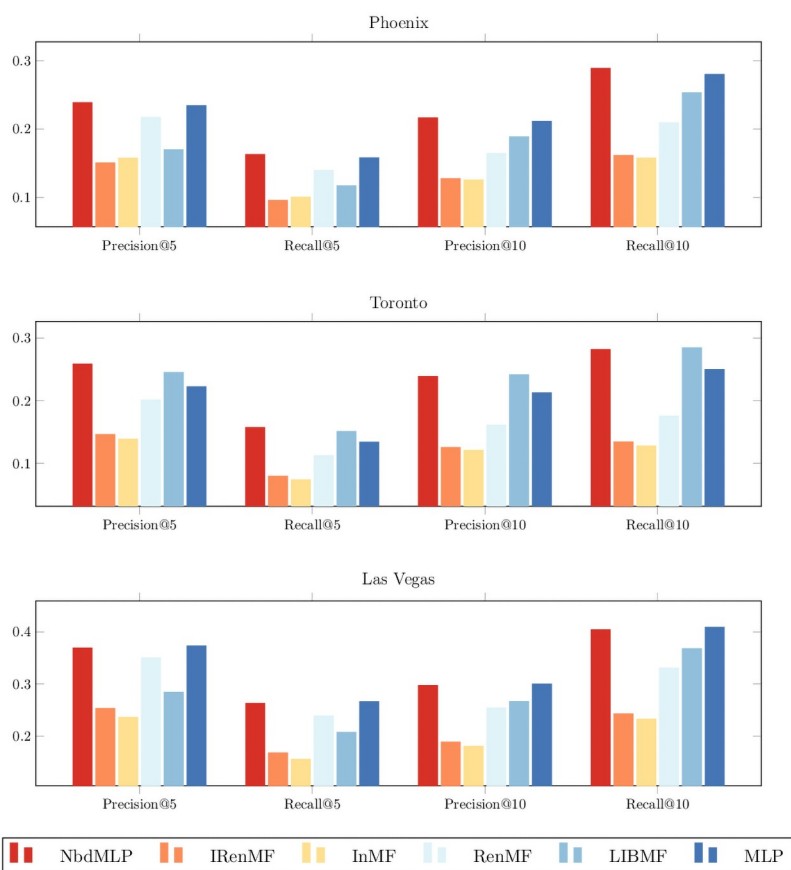

**Fig 3. The precision and recall of NbdMLP and five baselines.**

the small difference and NbdMLP's better performance with P@5 and R@5, we argue that NbdMLP outperforms matrix factorization–based baselines across three cities.

IRenMF, InMF, and RenMF also exploit neighborhood effect. The better performance of NbdMLP compared to these baselines indicates that integrating neighborhood features into a deep neural network is an effective way of building neighborhood-based POI recommenders.

The definition of neighborhood effect in InMF is very similar to the aggregated *personalized* features in Section 4. NbdMLP outperforms InMF with all metrics across three cities. For example, in terms of P@5, the improvement of NbdMLP over InMF is 51.64%, 86.47%, and 56.16% on Phoenix, Toronto, and Las Vegas, respectively. Even when we compare the performance of NbdMLP with the best performing neighborhood-based baseline, RenMF, in terms of P@5, NbdMLP yields improvements of 9.86%, 28.30%, and 5.33% on Phoenix, Toronto, and Las Vegas, respectively.

Interestingly, NbdMLP outperforms MLP with all metrics in Phoenix and Toronto, but not in Las Vegas. This observation suggests that neighborhood effect is not a major factor in deciding a user's POI's visiting behavior in Las Vegas. We conjecture that, in a tourism city, such as Las Vegas, a user's POI visiting behavior is driven mainly by the popularity of a POI, but not the nearby POIs.

## 6.5 Experiments on hidden layers

We experiment with different hidden layer configurations of NbdMLP. Since the number of neighborhood features can vary, we only indicate the size of hidden units of MLP layers in Table 5, excluding the neighborhood features. However, the experiment results are conducted with *property* features. For example, "32 → 16 → 8" indicates that the size of the concatenated user and POI embeddings is 32 (where the embedding size of users and POIs is 16 each), and the hidden units size of the second layer and the last layer is 16 and 8, respectively.

Table 5 only reports P@5 and R@5 for brevity. We highlight the highest value in each column. First, for a fixed embedding size, if we increase the size of hidden units of other layers (e.g., 32 → 16 → 8 vs. 32 → 32 → 16), both P@5 and R@5 improve across three cities. The results suggest that increasing the width of hidden layers improves performance.

If we only increase the depth of hidden layers, with the embedding size fixed (32 → 16 → 8 vs. 32 → 32 → 16 → 8), only Las Vegas shows improved P@5 and R@5. It suggests that increasing the depth of hidden layers may cause overfitting and degrade the performance, especially with a small data set. The same pattern remains if we increase the embedding size of users (POIs), but make the other hidden layers remain the same (32 → 32 → 16 vs. 64 → 32 → 16). Because Las Vegas has a much larger data set than Phoenix and Toronto, increasing the model complexity does improve the performance.

**Table 5. P@5 and R@5 for hidden layer configurations.**

| Hidden layers | Phoenix | | Toronto | | Las Vegas | |
|---|---|---|---|---|---|---|
| | **P@5** | **R@5** | **P@5** | **R@5** | **P@5** | **R@5** |
| 32 → 16 → 8 | 0.2262 | 0.1565 | 0.2529 | 0.1526 | 0.3557 | 0.2574 |
| 32 → 32 → 16 → 8 | 0.2179 | 0.1478 | 0.2563 | 0.1563 | 0.3674 | 0.2620 |
| 32 → 32 → 16 | **0.2387** | **0.1627** | **0.2586** | **0.1574** | 0.3688 | 0.2628 |
| 64 → 32 → 16 | 0.2277 | 0.1550 | 0.2392 | 0.1457 | 0.3724 | 0.2643 |
| 64 → 32 → 16 → 8 | 0.2142 | 0.1454 | 0.2348 | 0.1417 | **0.3836** | **0.2733** |

Overall, this experiment suggests that increasing the complexity of hidden layers does not always improve the performance. The performance depends on the scale and context of the data sets.

## 7 Conclusions

We propose different sets of features to quantify a user's preference for a neighborhood. These features constitute an interpretable framework for analyzing the neighborhood effect with respect to POI recommendation.

Our evaluation of neighborhood features indicates that *property* features (which capture the environmental context of a neighborhood) perform in par with *personalized* features (which capture a user's personalized preference over nearby POIs). In addition, there is no significant difference among aggregating functions of *personalized* features.

We propose NbdMLP, a deep neural network–based recommender, to exploit neighborhood features. We demonstrate the promise of building a deep learning–based recommender by incorporating heterogeneous information and conduct extensive experiments with different hidden layer parameters. NbdMLP's higher precision and recall over two matrix factorization models, specifically in cities representing daily life style, suggest the effectiveness of our approach in such cities. The influence of the neighborhood effect in tourism and possibly other types of cities remains to be studied.

## Author Contributions

**Conceptualization:** Pradeep K. Murukannaiah.

**Data curation:** Guangchao Yuan.

**Formal analysis:** Guangchao Yuan.

**Investigation:** Guangchao Yuan.

**Methodology:** Guangchao Yuan.

**Software:** Guangchao Yuan, Pradeep K. Murukannaiah.

**Writing – original draft:** Guangchao Yuan.

**Writing – review & editing:** Guangchao Yuan, Munindar P. Singh, Pradeep K. Murukannaiah.

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
