## [Decision Letter · Decision Letter 0]

28 Apr 2021

PONE-D-21-04507

An Interpretable Framework for Investigating Neighborhood Effect in POI Recommendation

PLOS ONE

Dear Dr. Murukannaiah,

Thank you for submitting your manuscript to PLOS ONE. After careful consideration, we feel that it has merit but does not fully meet PLOS ONE’s publication criteria as it currently stands. Therefore, we invite you to submit a revised version of the manuscript that addresses the points raised during the review process.

We look forward to receiving your revised manuscript.

Kind regards,

Haoran Xie

Academic Editor

PLOS ONE

Journal Requirements:

The authors have declared that no competing interests exist.

We note that one or more of the authors are employed by a commercial company: Microsoft

2a.              Please provide an amended Funding Statement declaring this commercial affiliation, as well as a statement regarding the Role of Funders in your study. If the funding organization did not play a role in the study design, data collection and analysis, decision to publish, or preparation of the manuscript and only provided financial support in the form of authors' salaries and/or research materials, please review your statements relating to the author contributions, and ensure you have specifically and accurately indicated the role(s) that these authors had in your study. You can update author roles in the Author Contributions section of the online submission form.

2b. Please also provide an updated Competing Interests Statement declaring this commercial affiliation along with any other relevant declarations relating to employment, consultancy, patents, products in development, or marketed products, etc. 

Reviewers' comments:

Reviewer's Responses to Questions

**Comments to the Author**

1. Is the manuscript technically sound, and do the data support the conclusions?

Reviewer #1: Yes

Reviewer #2: Yes

2. Has the statistical analysis been performed appropriately and rigorously? 

Reviewer #1: Yes

Reviewer #2: N/A

3. Have the authors made all data underlying the findings in their manuscript fully available?

Reviewer #1: Yes

Reviewer #2: Yes

4. Is the manuscript presented in an intelligible fashion and written in standard English?

Reviewer #1: Yes

Reviewer #2: Yes

5. Review Comments to the Author

Reviewer #1: The manuscript describes a new method to incorporate the neighborhood information when predicting a user's preference of a point-of-interest. The overall writing is fine and the authors have explained the functions used in different features.

Few comments are listed below:

1. A bit of confusing about the number of times a user visits a POI. At lines 148-149, the authors said that "... not the exact number if the user visited the POI more than once.", but in Table 2, n_{u,l} is the number of times a user u visits POI l. Please clarify.

2. Lines 160-162, the size of a neighborhood is the number of POIs within a distance, e.g. 100 m, 200 m, 500 m. This doesn't equal to the description at lines 161-162. There might be cases that the distance is larger, but no POIs are included.

3. Lines 202-204, is there any reason why a word has to be appeared at least four times in a city? For example, why not five times?

4. Line 421, is the method "IRenMF"? The subtitle is the same as line 430.

5. Lines 463-464, "... the embedding size of users (POIs) is 16 ...", should it be 32, as indicated at line 463?

6. Please indicate the meaning of bold text in tables, are they the best values in the corresponding categories?

Reviewer #2: This manuscript addresses a problem of recommending potential points of interest (POIs) to users. The particular emphasis is on a phenomenon called the "neighborhood effect" that relates to a user's preference for POIs in the neighborhood of a POI whose preference is already established. The authors accomplish two tasks. First, they identify features potentially related to the neighborhood effect and employ both logistic regression and FastTree to evaluate combinations of features. Second, they develop an artificial neural network (ANN) model meant to take advantage of the neighborhood effect and evaluate it against some established matrix factorization techniques and a simplified ANN. At a high level, the research seems good and the results correct, but I do have some detailed comments and thoughts to address.

First, I marked up some required changes in red on the manuscript and scanned the result to a PDF, which is attached. Note that "data set" is properly two words, though this is often abused.

Second, many of the references are on the older side; the manuscript would be improved by refreshing the References list with more recent papers.

Third, on page 5, the manuscript makes an offhand remark about a frequency distribution being a power law. This kind of casual remark is often made without justification, as is done here. It is likely the distribution is not a true power law in a statisitical sense, so it is best to delete this claim.

Fourth, there is little to no justification for the ANN architecture (Section 6.1); it is just stated as a fact. This approach is common in ANN research and does not reflect well on the area. Answer why you made the architectural decisions you did in a deeper way.

Fifth, the Problem Formulation (Section 3) is merely two sets of questions. There is no real statement of computer science problems, where the inputs and outputs (solutions) are mathematically specified so that the authors and the readers can know when a problem is solved. I note that the authors are all computer scientists, so they know how to mathematically specify computational problems. Such specifications should be developed in any revision.

Sixth and finally, I never saw a mathematical definition of the neighborhood effect, so it was never clear what it really is; this should be fixed in any revision.

6. PLOS authors have the option to publish the peer review history of their article (what does this mean?). If published, this will include your full peer review and any attached files.

Reviewer #1: No

Reviewer #2: No

---

## [Author Response · Author response to Decision Letter 0]

15 Jun 2021

Thanks to the editor and the reviewers for very helpful comments. We have addressed all the comments in this revision.

Our response to the review comments are included in a PDF file we are submitting as part of the revision.

In the revision workflow, we could not edit the information on Competing Interests and Funding Statement. Thus, we are including the pertinent information below.

The first author (Guangchao Yuan) is employed by Microsoft. Microsoft provided support in the form of salaries for Guangchao Yuan, but did not have any additional role in the study design, data collection and analysis, decision to publish, or preparation of the manuscript. The specific role of the author is articulated in the ‘author contributions’ section. This does not alter our adherence to PLOS ONE policies on sharing data and materials.

---

## [Decision Letter · Decision Letter 1]

22 Jul 2021

An Interpretable Framework for Investigating Neighborhood Effect in POI Recommendation

PONE-D-21-04507R1

Dear Dr. Murukannaiah,

We’re pleased to inform you that your manuscript has been judged scientifically suitable for publication and will be formally accepted for publication once it meets all outstanding technical requirements.

Kind regards,

Haoran Xie

Academic Editor

PLOS ONE

Additional Editor Comments (optional):

The paper is ready for publication.

Reviewers' comments:

Reviewer's Responses to Questions

**Comments to the Author**

1. If the authors have adequately addressed your comments raised in a previous round of review and you feel that this manuscript is now acceptable for publication, you may indicate that here to bypass the “Comments to the Author” section, enter your conflict of interest statement in the “Confidential to Editor” section, and submit your "Accept" recommendation.

Reviewer #1: All comments have been addressed

Reviewer #2: All comments have been addressed

2. Is the manuscript technically sound, and do the data support the conclusions?

Reviewer #1: (No Response)

Reviewer #2: Yes

3. Has the statistical analysis been performed appropriately and rigorously? 

Reviewer #1: (No Response)

Reviewer #2: N/A

4. Have the authors made all data underlying the findings in their manuscript fully available?

Reviewer #1: (No Response)

Reviewer #2: Yes

5. Is the manuscript presented in an intelligible fashion and written in standard English?

Reviewer #1: (No Response)

Reviewer #2: Yes

6. Review Comments to the Author

Reviewer #1: (No Response)

Reviewer #2: (No Response)

7. PLOS authors have the option to publish the peer review history of their article (what does this mean?). If published, this will include your full peer review and any attached files.

Reviewer #1: No

Reviewer #2: **Yes: **Lenwood S. Heath

---

## [Editor Report · Acceptance letter]

28 Jul 2021

PONE-D-21-04507R1 

An Interpretable Framework for Investigating the Neighborhood Effect in POI Recommendation 

Dear Dr. Murukannaiah:

I'm pleased to inform you that your manuscript has been deemed suitable for publication in PLOS ONE. Congratulations! Your manuscript is now with our production department. 

Kind regards, 

on behalf of

Professor Haoran Xie 

Academic Editor

PLOS ONE